# Electromagnetic Wavefront Engineering by Switchable and Multifunctional Kirigami Metasurfaces

**DOI:** 10.3390/nano15010061

**Published:** 2025-01-02

**Authors:** Yingying Wang, Yang Shi, Liangwei Li, Zhiyan Zhu, Muhan Liu, Xiangyu Jin, Haodong Li, Guobang Jiang, Jizhai Cui, Shaojie Ma, Qiong He, Lei Zhou, Shulin Sun

**Affiliations:** 1Shanghai Engineering Research Centre of Ultra Precision Optical Manufacturing, Department of Optical Science and Engineering, School of Information Science and Technology, Fudan University, Shanghai 200433, China; 21210720013@m.fudan.edu.cn (Y.W.); 22110720011@m.fudan.edu.cn (Y.S.); 21210720008@m.fudan.edu.cn (L.L.); 21110720021@m.fudan.edu.cn (Z.Z.); 21110720013@m.fudan.edu.cn (M.L.); 22210720009@m.fudan.edu.cn (X.J.); shaojiema@fudan.edu.cn (S.M.); 2Yiwu Research Institute, Fudan University, Chengbei Road, Yiwu 322000, China; 3State Key Laboratory of Surface Physics (Ministry of Education), Fudan University, Shanghai 200433, China; 19110190007@fudan.edu.cn (H.L.); qionghe@fudan.edu.cn (Q.H.); 4Key Laboratory of Micro and Nano Photonic Structures (Ministry of Education), Fudan University, Shanghai 200433, China; 5Department of Materials Science and State Key Laboratory of Molecular Engineering of Polymers, Fudan University, Shanghai 200438, China; 21110300010@m.fudan.edu.cn (G.J.); jzcui@fudan.edu.cn (J.C.); 6International Institute of Intelligent Nanorobots and Nanosystems, Fudan University, Shanghai 200438, China; 7Collaborative Innovation Center of Advanced Microstructures, Nanjing 210093, China

**Keywords:** metasurface, kirigami technique, switchable functionality, anomalous reflection

## Abstract

Developing switchable and multifunctional metasurfaces is essential for high-integration photonics. However, most previous studies encountered challenges such as limited degrees of freedom, simple tuning of predefined functionality, and complicated control systems. Here, we develop a general strategy to construct switchable and multifunctional metasurfaces. Two spin-modulated wave-controls are enabled by the proposed high-efficiency metasurface, which is designed using both resonant and geometric phases. Furthermore, the switchable wavefront tailoring can also be achieved by flexibly altering the lattice constant and reforming the phase retardation of the metasurfaces based on the “rotating square” (RS) kirigami technique. As a proof of concept, a kirigami metasurface is designed that successfully demonstrates dynamic controls of three-channel beam steering. In addition, another kirigami metasurface is built for realizing tri-channel complex wavefront engineering, including straight beam focusing, tilted beam focusing, and anomalous reflection. By altering the polarization of input waves as well as transformation states, the functionality of the metadevice can be switched flexibly among three different channels. Microwave experiments show good agreement with full-wave simulations, clearly demonstrating the performance of the metadevices. This strategy exhibits advantages such as flexible control, low cost, and multiple and switchable functionalities, providing a new pathway for achieving switchable wavefront engineering.

## 1. Introduction

Creating multiple functionalities in a single device is highly demanded for the development of high-integration photonics. However, traditional elements, including refractive devices (e.g., lens) and diffraction devices (e.g., grating), usually suffer from problems such as limited functionality, bulky configuration, and low performance. Recently, great advancements in metasurfaces have unlocked new possibilities for achieving high-integration and diverse manipulations of electromagnetic (EM) waves at a subwavelength scale. Metasurfaces, a type of quasi-two-dimensional metamaterial composed of subwavelength microstructures with spatially modulated responses, have exhibited extraordinary capabilities to control EM or light waves. Researchers have realized numerous fascinating effects for tailoring wavefronts, including anomalous beam deflection [1,2], surface wave excitations [3,4], photonic spin Hall effect (PSHE) [5,6,7], beam focusing [8,9], and holographic imaging [10,11], among others. Although these metadevices enable the promising potential of controlling EM fields in the desired manner, they usually face challenges such as low efficiency and lack of degrees of freedom. In particular, most devices exhibit fixed functionalities after fabrication, reducing the flexibility of their practical applications.

Recently, tunable metasurfaces composed of building blocks combined with active components or materials offer great potential for dynamically modulating EM or light waves. For instance, researchers have utilized semiconductors [12], phase change materials [13,14,15], or 2D materials [16,17,18,19] to effectively modulate the scattering field amplitude or phase of meta-atoms by altering their material properties in response to an applied electrical [16,18], thermal [14,19], optical [20,21], or chemical [22,23] stimulus. While most tuning strategies are utilized for the flexible change of the fundamental EM parameters based on homogeneous metamaterials, they are usually limited by the tuning range or specific operating frequency bands. Typically, via integrating electronic components such as PIN diodes and varactors with the local meta-atoms, researchers have constructed homogeneous or gradient metasurfaces for achieving dynamic wavefront engineering, including arbitrary polarization conversion [24,25], beam steering [26,27,28,29,30,31,32], active EM cloaking [33], and dynamic holography [34]. These methods show the advantages of quick response and multiple functionalities, but still suffer from the problems of complicated control systems and high cost. Therefore, a flexible control and low-cost approach to achieve tunable/switchable wavefront control is highly desired.

Another approach to achieve dynamic wavefront modulation relies on altering the spatial arrangement of the meta-atoms. For instance, researchers adopt microelectromechanical systems (MEMS) to alter the relative displacement [35,36] or spatial rotation angle [37,38] between multilayer metasurfaces, enabling applications such as focal length tuning and dynamic beam shaping. On the other hand, people utilize elastic materials [39,40,41] or origami/kirigami substrates [42,43,44,45,46] to change the relative spacing between the meta-atoms within the metasurfaces, achieving the dynamical control of transmission characteristics, focal length, and holographic image. In principle, such a strategy, based on the global tuning of lattice constant, is applicable to a wide range of frequency bands and material systems. However, previous mechanical systems usually lacked the degree of freedom to control the local optical properties of meta-atoms, which significantly restricts their applications based on multi-functional wavefront reconstruction.

In this paper, we propose a switchable metasurface based on the “rotating square” (RS) kirigami technique to achieve three different functionalities. Utilizing the flexible spatial rotation through the RS kirigami technique [47], we can simultaneously tailor the local phase retardation and global lattice constant, achieving switchable wavefront manipulation. Moreover, the proposed metasurfaces, consisting of a series of elaborately designed microstructures behaving like effective half-wave plates, are encoded with both the resonant phase and Pancharatnam–Berry (PB) phase [5,48], further enriching their functionalities through the additional degree of freedom, i.e., input polarization. To validate our idea, we employ the RS kirigami technique to design the switchable multi-functional metasurfaces. The first metadevice is demonstrated to achieve the tri-channel beam bending effect via tuning the rotation state and input spin. Another switchable metasurface is further constructed for realizing more complicated wavefront engineering, including beam deflection and focusing (as shown in Figure 1). Microwave experiments, as well as full-wave simulations, perfectly demonstrate the theoretical predictions. Our scheme exhibits the advantages of switchable control, multiple functionalities, and flexible implementation, which are highly demanded for multifunctional switchable wavefront engineering.

## 2. Results and Discussions

### 2.1. Basic Concept

We first introduce the fundamental principle and working process of our switchable kirigami metasurface for achieving multifunctional wavefront engineering, as schematically shown in Figure 1. In the initial undeformed state (β = 0°), the metasurface encoded simultaneously with both the resonant phase and PB phase can realize spin-delinked anomalous dual-functionalities (i.e., target 1 and target 2) under the illumination of left circular polarization (LCP) and right circular polarization (RCP) waves [49,50,51]. Based on the RS kirigami technique, a global tensile force applied to the transformable substrates underneath the metadevice will cause the adjacent meta-atoms to rotate in opposite directions, as shown in Figure 1 [47]. During the transformation process, the meta-atoms located in the gray regions will rotate counter-clockwise, while those in the black regions will rotate clockwise. Such a rotation operation can modulate the orientations of the meta-atoms, thereby simultaneously altering the encoded phase profile and the lattice constant of the meta-atoms. We use rotation angle β, defined as the intersection angle between the bottom edge of meta-atoms and the x-axis of the global laboratory coordinates, to describe the degree of rotation transformation of the RS kirigami metasurface (see Figure 1). Generally, when the metasurface is transformed to an arbitrary state of β, the change of PB phase carried by the meta-atoms can be described by:(1)$${∆\phi }_{PB}^{\beta }=({-1)}^{i+j}\sigma 2\beta$$
and the lattice constant is also accordingly varied:(2)$$p^{\beta }=p^{0}[\sin\left(\beta \right)+cos(\beta )]$$
where i and j represent the sequence number of meta-atoms in the x and y directions, respectively, σ represents the spin state of the incident CP light (σ=+1: LCP; and σ=−1: RCP), and $p^{0}$ represents the lattice constant of meta-atom in the transformation state of β=0°. Therefore, by changing the rotation angle β, we can reconfigure the phase profile of the metasurface, thereby modulating its functionality. The design details of the RS transformable substrate based on 3D printing techniques are discussed in Appendix A.

According to the Jones’ matrix analysis, while the generic reflective meta-atoms are illuminated by the CP light, the reflection coefficient can be usually decomposed into two different components [6,7]. One is the spin-conserved normal mode described by ${\tilde{r}}_{n}=\frac{1}{2}\left(r_{uu}+r_{vv}\right)$ and the other one is a spin-flipped anomalous mode described by ${\tilde{r}}_{a}=\frac{1}{2}{\left(r_{uu}-r_{vv}\right)e}^{i\phi_{PB}}$, in which $\left\{u,v\right\}$ are the two principal axes of the meta-atoms and $\phi_{PB}=2\sigma \theta$ is the so-called PB phase dependent on the input spin state and orientation (θ) of the meta-atoms. We note that the two spin-flipped anomalous modes can be utilized for wavefront engineering, which is further tailored by the rotation operation. To achieve the high-efficiency wavefront control, we optimize the design of the meta-atoms to suppress the normal mode, which requires the criterion of $r_{uu}+r_{vv}=0$. If the material losses can be neglected, the amplitudes of two co-polarization reflection coefficients for the meta-atoms in total reflection configuration, i.e.,$\left|r_{uu}\right|$ and $\left|r_{vv}\right|$, are both 100%. Therefore, the aforementioned criterion will be changed to an effective new form $\left|\phi_{uu}-\phi_{vv}\right|=\pi$, suggesting that the ideal meta-atoms should exhibit half-wave plate behavior.

To achieve spin-modulated multiple functionalities, the composite-phase metasurface is designed to employ two different kinds of phase-control mechanisms. If rewriting the reflection coefficient of the anomalous mode as ${\tilde{r}}_{a}=\frac{1}{2}{(r_{uu}-r_{vv})e}^{i\phi_{PB}}=\left|{\tilde{r}}_{a}\right|e^{i\left(\phi_{PB}+\phi_{res}\right)}$, we can derive the so-called resonant phase $\phi_{res}=arg\left(\frac{r_{uu}-r_{vv}}{2}\right)$, which is determined by the geometric parameters of meta-atoms [42]. For LCP and RCP illumination cases, two independent phase profiles ($\phi_{LL}and\phi_{RR}$), combining both the spin-dependent PB phase and spin-independent resonant phase, can be integrated into the anomalous reflection modes of the composite phase metasurface:(3)$$\left\{\begin{array}{c} \phi_{LL}\left(x,y\right)=\phi_{res}\left(x,y\right)+\phi_{PB}\left(x,y\right)\\ \phi_{RR}\left(x,y\right)=\phi_{res}\left(x,y\right)-\phi_{PB}\left(x,y\right) \end{array}\right.$$

Here, we need to emphasize that, considering that the direction of the input beam is reversed after reflection, the spin state of the anomalous reflection beam is the same as that of the incident beam. At the initial state of β=0°, the composite-phase metasurface composed by the 100% polarization conversion ratio (PCR) meta-atoms will deflect the input LCP or RCP wave to two distinct anomalous reflection channels, achieving the high-efficiency bifunctional wavefront manipulations (denoted as target 1 and target 2) as depicted in Figure 1a. At the transformed states, the metasurface will also release the third spin-conserved normal reflection mode, exhibiting a polarization-independent phase response:(4)$$\phi_{RL}\left(x,y\right)=\phi_{LR}\left(x,y\right)=\phi_{res}\left(x,y\right)$$

During the transformation process, the meta-atoms located in the black and gray regions rotate in opposite directions, resulting in a phase difference between them. Therefore, the desired constructive interference condition of the global phase profile will gradually fail, suppressing the anomalous mode and releasing the normal mode. As shown in Figure 1b, tri-channels, denoted as targets 1–3, are available in such a metasurface for multifunctional wavefront controls. In particular, in the state of β=45°, the orientations of two neighboring meta-atoms are changed by +45° and −45°, respectively. It implies that the phase difference between them equals to 180°, corresponding to the destructive interference condition. Consequently, two anomalous modes are terminated, leaving only the single normal mode, which corresponds to the case of the third functionality denoted as target 3 in Figure 1c. In summary, we can realize the modulation among three different functionalities by controlling the transformation state of the kirigami metasurface and the illumination of different CP waves.

To verify our concept, we start to design high-efficiency meta-atoms as the building block of the desired metadevice operating in the microwave regime. In a total reflection configuration, we propose a metal-insulator-metal (MIM) typed microstructure satisfying the mirror symmetry, which consists of a curved H-shaped microstructure as the top layer, a 3 mm-thick F4B dielectric layer ($\varepsilon_{r}=3.5$) in the middle, and a continuous copper film as the bottom layer. Numerical optimization is adopted to determine the structural parameters of the desired meta-atoms that satisfy the aforementioned theoretical criterion. Figure 2a shows one example of such meta-atoms in a periodic array with the following structural parameters: r = 3 mm, w = 0.85 mm, α = 120°, and p = 8 mm. Figure 2b illustrates the spectra of the reflection phase and PCR of such meta-atoms obtained by numerical simulations and experimental measurements. It is noted that the phase difference between $\phi_{uu}$ and $\phi_{vv}$ can be maintained around 180° within the whole frequency window. Here, the PCR of the meta-atoms, defined as ${\left|(r_{uu}-r_{vv})/2\right|}^{2}$ according to the previous Jones’ matrix analysis, reaches approximately 100% in the frequency range of 9~12 GHz. The measurement results are in good agreement with the simulation results, verifying the high performance of such a metadevice. Furthermore, as the opening angle α of the curved H-shaped microstructure increases, the resonant phase of such meta-atoms can be continuously adjusted within a large modulation range, as shown in Figure 2c. Meanwhile, PCRs of the meta-atoms with different α still remain nearly 100% across almost the entire bandwidth, as shown in Figure 2d. These results indicate that we have built up the database of the high-performance composite meta-atoms for constructing the desired spin-delinked bifunctional metasurface according to Equation (3).

### 2.2. Switchable Multiple-Beam Meta-Reflector

We first designed the kirigami metasurface to achieve the switchable multiple reflection beam manipulations as the first example. Such a metasurface consists of 29 × 29 elements with a total size of $238\times 238{mm}^{2}$ in the initial state (β = 0°). According to our design, such a spin-delinked metasurface will deflect incident LCP and RCP beams towards the directions of $\theta_{r}^{LL}=-30{}^{\circ}$ and $\theta_{r}^{RR}=45{}^{\circ}$ at 10 GHz, respectively. The phase profiles carried by three channels of the metasurface illuminated by the normally incident CP wave can be expressed as:(5)$$\left\{\begin{array}{c} \phi_{LL}\left(x,y\right)=k_{0}\sin\left(\theta_{r}^{LL}\right)x\\ \phi_{RR}\left(x,y\right)=k_{0}\sin\left(\theta_{r}^{RR}\right)x\\ \phi_{RL}\left(x,y\right)=\phi_{LR}\left(x,y\right)=\left[k_{0}\sin\left(\theta_{r}^{LL}\right)x+k_{0}\sin\left(\theta_{r}^{RR}\right)x\right]/2 \end{array}\right.$$
where $k_{0}$ is the wavevector of the EM wave in free space. According to Equation (3) and Equation (5), we can obtain the distributions of the desired resonant phase $\phi_{res}$ and PB phase $\phi_{PB}$, respectively, encoded by the meta-atoms at each local position of (x, y). Finally, we can determine the full parameters, including opening angle α(x, y) and orientation angle θ(x,y), of each meta-atom on the composite-phase metasurface based on the relationship of $\alpha \sim \phi_{res}$ and $\theta \sim \phi_{PB}$ (see Appendix A of Appendix A). The reflection phase distributions of the metasurface for two anomalous modes and one normal mode based on the Equation (5) are shown in Figure 3d. Due to the high PCRs of the adopted meta-atoms, the normal mode of the metasurface at the initial state (β=0°) is suppressed. Therefore, the incident LCP and RCP waves are almost completely deflected into the two pre-designed anomalous reflection channels, as shown in Figure 3a.

As the kirigami metasurface transforms to a new state by force, it will gradually produce the normal reflection mode as a new channel (i.e., target 3), as schematically shown in Figure 3b. The adjacent meta-atoms will rotate in the opposite directions as described by Equation (1). Although the phase profiles encoded by the even and odd rows of meta-atoms still possess the same slope or gradient, they will exhibit a phase difference of σ4β and thus no longer satisfy the strict in-phase condition, as depicted in Figure 3e. The consequence is that the anomalous modes are suppressed and the normal mode is released.

Intriguingly, the kirigami metasurface will perfectly match the destructive interference condition for the anomalous reflection channels while transformed to the state of β = 45°, thus generating only the single normal mode as shown in Figure 3c. For such a special case, the orientations of adjacent meta-atoms change by ±45°, resulting in a phase difference of 180° for the two anomalous modes. Consequently, under the illumination of the CP wave, anomalous reflection modes emitted by the odd and even rows of meta-atoms will completely cancel each other out, and only the spin-independent normal mode still survives. Such a phenomenon can be also attributed to the nearly-zero effective PCR of the composite meta-atoms. While β is equal to 45°, the adjacent meta-atoms at even and odd rows will be perpendicular to each other, which can be treated as a single meta-molecule without EM anisotropy. Consequently, the effective PCR of such meta-molecule becomes 0, leading to the response of single normal mode generation for the metasurface. Additionally, during the transformation process, since the lattice constant of the meta-atoms will be varied simultaneously according to Equation (2), the reflection angles of both the normal and anomalous reflection modes will also undergo continuous variation.

Based on these analyses, we designed and fabricated the kirigami switchable metasurface that is attached on the resin transformation substrates prepared using 3D printing techniques. We adopt full-wave simulations and angle-resolved far-field experiments to verify the switchable functionalities of the metadevice in three representative states, including β = 0°, β = 22.5°, and β = 45°. During the measurement, we adopt the LCP or RCP horn antenna as the source to normally illuminate the kirigami metasurface, and then utilize another LCP or RCP horn antenna placed on a circular track with a radius of 1.2 m to detect the angular distribution of scattering electric field intensity with the co- or cross-polarization. Both CP horn antennas are connected to a vector network analyzer (Agilent E8362CPNA). To clarify the performance of the device, we normalize the scattering field intensity to that of the reference signal, which is the intensity of the specular beam reflected by a flat metallic mirror of the same size under identical illumination conditions. The efficiency of the metadevice can be obtained by dividing the integrated power carried by the desired reflection mode to that of the reference signal. When the kirigami metasurface is in the initial state (β = 0°), the normalized scattering field distributions at 10 GHz are depicted in Figure 3g,j, obtained by both simulations and experiments. The results demonstrate that only the two spin-modulated anomalous reflection modes exist and that the spin-independent normal mode is completely suppressed, regardless of the incident LCP or RCP wave. Figure 3h,k illustrate the angular distribution of the normalized scattering field of the kirigami metasurface transformed to the state of β = 22.5°, clarifying the co-existence of both two anomalous modes and one normal mode. Eventually, in the state of β = 45°, the kirigami metasurface exhibits the functionality of single normal mode generation independent on the spin state of input CP waves. Interestingly, since the resonant phase profile responsible for generating normal mode shows a discontinuous linear slope, as shown in Figure 3f, the metadevice will generate a co-polarization nonspecular deflection beam. Meanwhile, Figure 3i,l shows that a reflection signal along 0° direction coexists, which is caused by the discontinuity of such resonant phase profile covering only a limited range of about 180° (see Appendix A of Appendix A). This problem can be resolved by redesigning a new class of meta-atoms that can possess both high PCRs and resonant phases, covering the full range of 360°. All measurement results are consistent with the simulation results, perfectly verifying our theoretical predictions. In addition, because air gaps will appear between the adjacent meta-atoms of the kirigami metasurface during the transformation process, some of the energy will be transferred to the transmission channel, reducing the efficiency of the desired reflection modes.

Indeed, the proposed kirigami metasurface can achieve the switchable beam steering within a broad frequency band. Figure 4 shows the normalized scattering electric field intensity as the functions of working frequency and reflection angle $\theta_{r}$ for the metasurface in three transformation states, obtained through microwave measurements. At a broad frequency band from 9 to 12 GHz, the kirigami metasurface experiences dramatic changes in the achieved beam-steering functionalities in three transformation states, including dual anomalous-modes reflection, tri-modes generation, and the single normal-mode reflection, which are consistent with theoretical predictions. Full-wave simulations are also adopted to confirm such broadband performance (see Appendix A–S6 of the Appendix A), which exhibits good agreement with the measurement results.

Furthermore, we also demonstrate the tunable anomalous beam deflection effect by the kirigami metasurface during the full RS transformation process from β = 0° to 90°, as shown in Appendix A. While β increases from 45° to 90°, the kirigami metasurface exhibits the same lattice constant and phase gradient as compared to the cases of β varying from 45° to 0°. For example, we have simulated the beam bending effect for the kirigami metasurface at β = 67.5° and β = 90° (see Appendix A of Appendix A), which exhibit identical functionalities to those of the metasurface at β = 22.5° (Figure 3b) and β = 0° (Figure 3a). Therefore, the functionalities exhibit a symmetric variation trend as β varies from 0° to 90°. As shown in Appendix A of Appendix A, both the full-wave simulations and microwave experiments agree well with theoretical predictions.

### 2.3. Kirigami Metadevice for Achieving Switchable Complex Wavefront Engineering

We designed another kirigami metasurface to further validate their flexible and diverse functionalities for tailoring the EM wavefronts. The metasurface is created based on the same database of the meta-atoms discussed previously, enabling the anomalous reflection (target 1) for incident RCP wave and a three-dimensional beam focusing effect (target 2) for incident LCP wave, as shown in Figure 5a. To achieve the two spin-delinked anomalous modes, the desired phase profiles encoded by the composited phase metasurface at the target frequency of 10 GHz can be expressed as:(6)$$\left\{\begin{array}{c} \phi_{RR}\left(x,y\right)=k_{0}\sin\left(\theta_{r}^{RR}\right)x\\ \phi_{LL}\left(x,y\right)=k_{0}(\sqrt{F^{2}+\left(x^{2}+y^{2}\right)}-F) \end{array}\right.$$
where $\theta_{r}^{RR}=30{}^{\circ}$ is the corresponding anomalous reflection angle for RCP illumination, and F = 85 mm is the target focal length for LCP illumination. According to Equations (3) and (6) and the relationship of $\alpha \sim \phi_{res}$ and $\theta \sim \phi_{PB}$, we can obtain the distributions of the two characteristic parameters (θ, α) of the meta-atoms at the position of (x, y) (see Appendix A, Appendix A). Such device consists of 29 × 29 elements with a total size of $238\times 238{mm}^{2}$ in the initial state (β = 0°). Figure 5d,g depict the distributions of $\phi_{RR}\left(x,y\right)$ and $\phi_{LL}\left(x,y\right)$ in the undeployed state (β = 0°), showing linear and parabolic profiles, respectively. As the kirigami metasurface transforms to the states of β = 22.5° and β = 45°, the encoded phases $\phi_{RR}(x,y)$ and $\phi_{LL}\left(x,y\right)$ can be degenerated into two phase planes with the same profiles but a constant phase difference of 90° and 180°, respectively, as shown in Figure 5e,h and Figure 5f,i. According to the previous discussion, the interference effect between them suppresses the anomalous modes and releases the normal mode, as schematically shown in Figure 5b. Since only the resonant phase distribution ($\phi_{res}\left(x,y\right)=[k_{0}\sin\left(\theta_{r}^{RR}\right)x+k_{0}(\sqrt{F^{2}+\left(x^{2}+y^{2}\right)}-F)]/2$) is responsible for generating the spin-independent normal mode, and thus is immune to the variation of meta-atoms’ orientation during the transformation process. In the special state of β = 45°, the metadevice only permits the existence of the normal mode, which corresponds to the tilted beam focusing effect as shown in Figure 5c, because of the destructive interference condition for the anomalous modes.

We adopt both far-field and near-field measurements to verify the multi-functional and switchable wavefront controls by the fabricated metasurface in different transformation states (see the transformation process in Appendix A). Figure 6a–c show the normalized scattering field distribution of anomalous reflection mode (target 1) of such a metadevice in three states illuminated by RCP waves, based on the previous angle-resolved far-field measurement technique. We can see that the anomalous reflection mode becomes weak and eventually disappears with the increase of β, matching well with our theoretical predictions. Meanwhile, the anomalous reflection angle accordingly decreases due to the increase of lattice constant according to Equation (2). In the near-field measurement, a horn antenna is used to vertically illuminate the CP beam on the metasurface, and a monopole antenna is moved to scan the different electric field components (including their amplitude and phase) in the XOZ plane. Based on these data, we can retrieve the electric field distributions of LCP and RCP components $(denoted\;as\;E_{LCP}\;and\;E_{RCP}$) on the metadevice illuminated by LCP beam, as depicted in Figure 6d,f. As β increases from 0° to 45°, the intensity of anomalous mode for beam focusing (target 2) gradually decreases to zero, accompanied by an increase in focal length due to the change of lattice constant. We adopt the same strategy to illustrate the electric field pattern for the co-polarization normal mode (target 3) generated by the composite-phase metasurface, as shown in Figure 6g–i. It is noted that the third functionality of the tilted beam focusing effect gradually becomes strong and then dominates within three channels. At the states of β = 22.5° and β = 45°, the tilted focal point appears at (−90 mm, 141 mm) and (−95 mm, 178 mm), respectively. The corresponding simulation results of the electric field distributions for targets 2 and 3 are shown in Appendix A of Appendix A, which agree well with the measurement results.

We finally investigate the evolution of two beam focusing functionalities (i.e., Focus 1: straight focusing contributed by anomalous mode; Focus 2: tilted focusing contributed by normal mode) based on the kirigami metasurfce at different states illuminated by the LCP beam, as shown in Figure 7. When the metasurface is transformed to an arbitrary state of β, the focal length caused by the change of lattice constant is [39,52]:(7)$$\left\{\begin{array}{c} F_{z}^{\beta }=[\sin\left(\beta \right)+\cos\left(\beta \right){]}^{2}F_{z}^{0}\\ F_{x}^{\beta }=\left[\sin\left(\beta \right)+\cos\left(\beta \right)\right]F_{x}^{0} \end{array}\right.$$
where $F_{z}^{0}$ and $F_{x}^{0}$ are the focal lengths of the metasurface in the transformation state of β=0°. Figure 7b shows that the focal length of the Focus 1 increases continuously as β varies from 0° to 45°. The functionality of Focus 1, based on anomalous reflection mode, is completely suppressed in the β = 45°state. As β varies, the focal point of tilted Focus 2 originating from the normal reflection modes shows a continuous modulation trend along both the x and y directions, as shown in Figure 7c,d. For the case of β = 0°, the efficiency of the Focus 2 functionality achieved by such a metasurface is nearly zero. Both experimental measurements and full-wave simulations show consistency with theoretical analysis, confirming the multifunctional and switchable wave modulations based on the proposed RS kirigami metadevices.

## 3. Conclusions

We propose a novel scheme to construct switchable and multifunctional metasurfaces based on RS kirigami technology and the spin-delinked composite-phase coding strategy. Different from previous origami/kirigami techniques, this scheme enables the flexible modulations of the lattice constant as well as the phase profile of the kirigami metasurface. Meanwhile, the functionalities of such metadevices can be arbitrarily switched between the spin-modulated anomalous dual-channels and the spin-independent normal single-channel by altering the interference condition or the effective PCRs of the composite meta-atoms during the transformation process. As a proof of concept, we designed and fabricated the first RS kirigami metasurface for achieving the dynamical multi-channel beam steering. During the transformation process, the channel numbers and the deflection angles of the reflection modes are flexibly modulated. The proposed concept is quite general in that it can offer potential applications in switchable complex EM controls. Specifically, we designed another tri-functional metasurface for achieving straight beam focusing, tilted beam focusing, and anomalous reflection. The focal points or the deflection angles can be flexibly modulated by tuning the transformation state of the metadevice. The RS kirigami metasurfaces exhibit the advantages of flexible control, low cost, and multiple and switchable functionalities. Our strategy opens a new pathway for achieving switchable wavefront engineering based on multiple degrees of freedom. With the advancements in micro- and nano-fabrication techniques [53,54,55], we believe that this switchable kirigami technology can be applied to higher frequencies.

## Figures and Tables

**Figure 1 nanomaterials-15-00061-f001:**
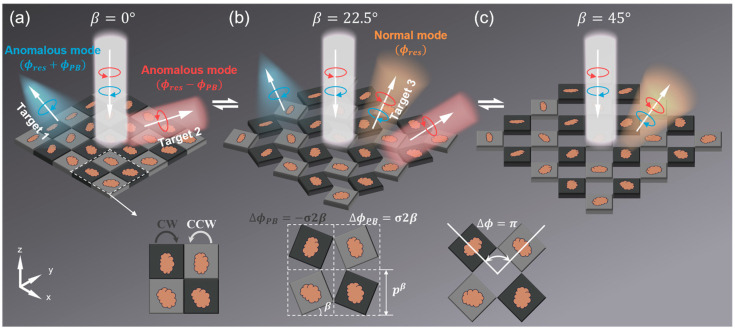
Schematics of the switchable gradient metasurface based on the RS kirigami technique. The adjacent meta-atoms of the kirigami metasurface undergo reverse rotations by the stretching force. During the transformation, the meta-atoms located in the black and gray panels will rotate clockwise (CW) and counter-clockwise (CCW), respectively. In this process, not only the local lattice constant of the meta-atoms is changed but also the global phase profile of the spin-flipped anomalous modes is reformed. By altering the transformation state of the kirigami metasurface, the achieved functionalities can be flexibly switched, e.g., dual-channels of two spin-modulated anomalous modes at β = 0° (**a**), tri-channels of two anomalous modes and one normal mode at β = 22.5° (**b**), and the single-channel response of only the normal mode at β = 45° (**c**). Here, β is utilized to characterize the transformation state of the metasurface, which is defined as the intersection angle between the bottom edge of meta-atoms and the x-axis.

**Figure 2 nanomaterials-15-00061-f002:**
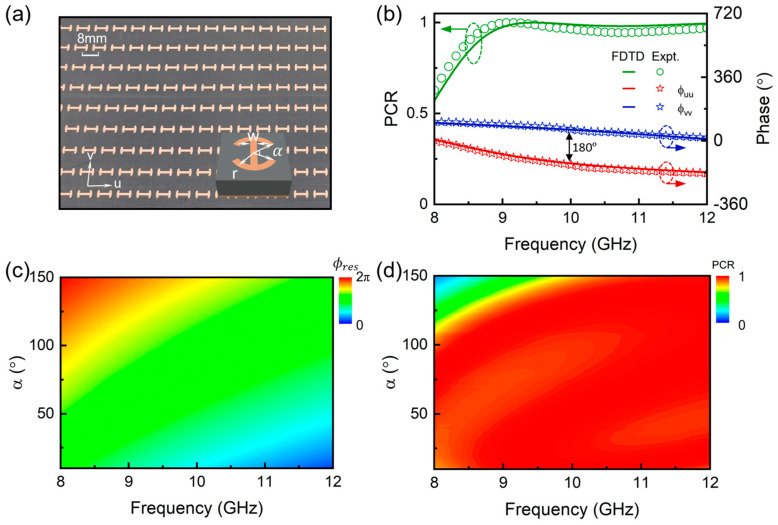
Design of high-efficiency composite-phase meta-atoms for the desired metasurface. (**a**) Schematic of the meta-atom array designed in the MIM configuration with a period of 8 mm. The thicknesses of the metal layer and dielectric film are 0.035 mm and 3 mm, respectively. The other parameters are listed as follows: r = 3 mm, w = 0.85 mm, and α = 120°. The opening angle α is a controllable parameter for adjusting the resonant phase of the meta-atoms. (**b**) Simulated and measured PCR and reflection phase spectra of the sample shown in (**a**) under the illumination of u-polarized and v-polarized EM waves, respectively. (**c**,**d**) Simulated resonant phase (**c**) and PCR (**d**) of the proposed meta-atom array as the functions of the opening angle α and frequency.

**Figure 3 nanomaterials-15-00061-f003:**
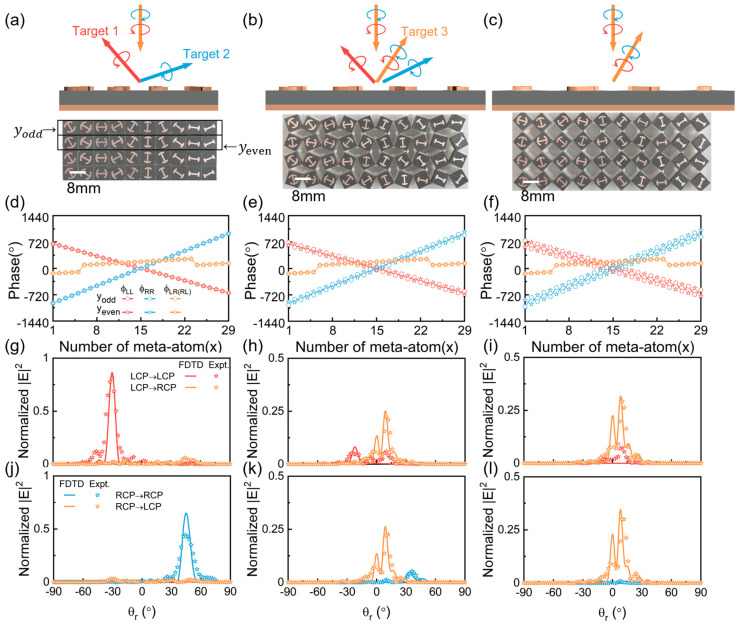
Characterization of the switchable multiple-beam meta-reflector based on the RS kirigami technique. (**a**–**c**) Schematical illustrations and sample images of the metasurface in β = 0°, β = 22.5°, and β = 45° states under the illumination of LCP and RCP waves. (**d**–**f**) The reflection phase profiles of the kirigami metasurfaces in three states illuminated by the LCP and RCP waves at the frequency of 10 GHz. (**g**–**i**) Measured (symbols) and simulated (lines) normalized scattering field angular distributions of the kirigami metasurfaces in three states illuminated by LCP wave at 10 GHz. (**j**–**l**) Measured (symbols) and simulated (lines) normalized scattering field angular distributions of kirigami metasurfaces in three states illuminated by RCP wave at 10 GHz.

**Figure 4 nanomaterials-15-00061-f004:**
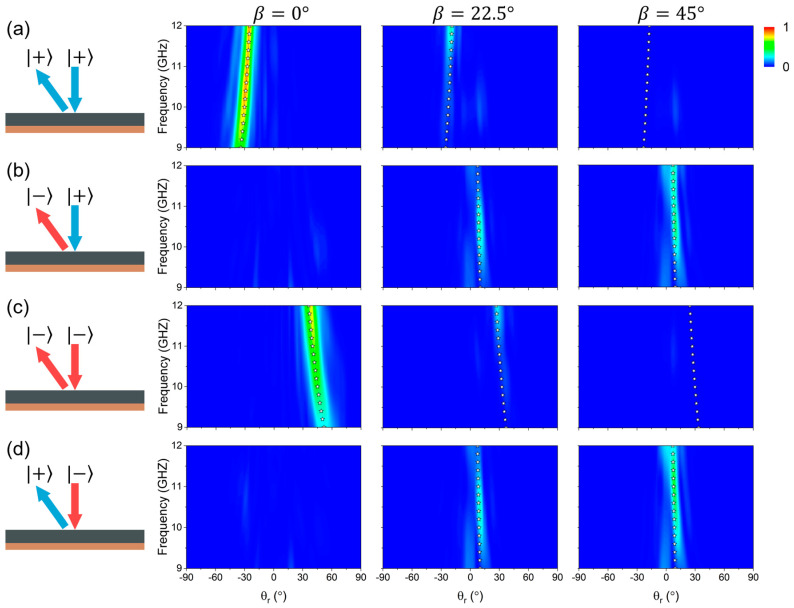
Experimental verification of the broadband switchable multiple beam deflection by RS kirigami metasurface. (**a**–**d**) Measured normalized scattering field intensities with LCP ($\left|\left.+\right\rangle \right)$
and RCP ($\left|\left.-\right\rangle \right)$ as the functions of working frequency and reflection angle for the kirigami metasurfaces in three states illuminated by normally incident LCP or RCP wave. Here, open stars represent the positions predicted by the generalized Snell’s law.

**Figure 5 nanomaterials-15-00061-f005:**
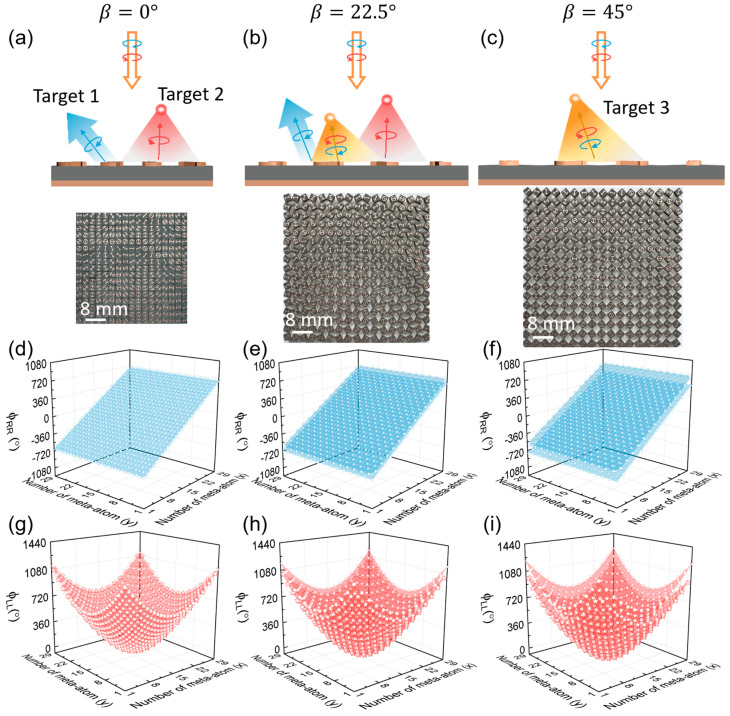
Switchable and multifunctional complex wavefront engineering based on the RS kirigami technique. (**a**–**c**) Schematical illustrations and sample images of the kirigami metasurface in β = 0°, β = 22.5°, and β = 45° states under LCP and RCP wave illumination. (**d**–**f**) The reflection phase distributions $\phi_{RR}(x,y)$ of kirigami metasurfaces in three states illuminated by the RCP wave at the frequency of 10 GHz. (**g**–**i**) The reflection phase distributions $\phi_{LL}(x,y)$ of kirigami metasurfaces in three states illuminated by the LCP wave at the frequency of 10 GHz.

**Figure 6 nanomaterials-15-00061-f006:**
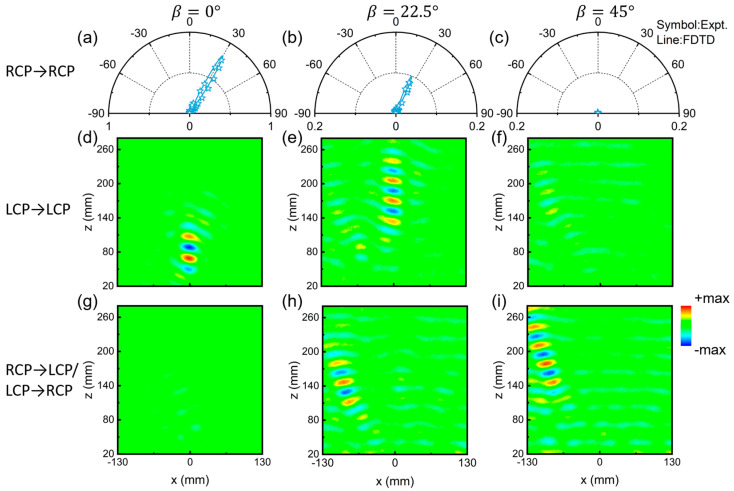
Characterization of the switchable and multifunctional complex wavefront engineering based on the RS kirigami technique. (**a**–**c**) Measured (symbols) and simulated (lines) normalized scattering field angular distributions with RCP of the kirigami metasurfaces in three states of β = 0°, β = 22.5°, and β = 45° illuminated by normally incident RCP wave. (**d**–**f**) Measured electric field distribution with LCP $\left(E_{LCP}\right)$
in the XOZ plane for the kirigami metasurfaces in three states illuminated by normally incident LCP wave. (**g**–**i**) Measured electric field intensity distribution of the normal mode ($E_{LCP}$ or $E_{RCP})$ in the XOZ plane for the kirigami metasurfaces in three states illuminated by normally incident LCP or RCP wave. Here, the frequency is fixed at 10 GHz.

**Figure 7 nanomaterials-15-00061-f007:**
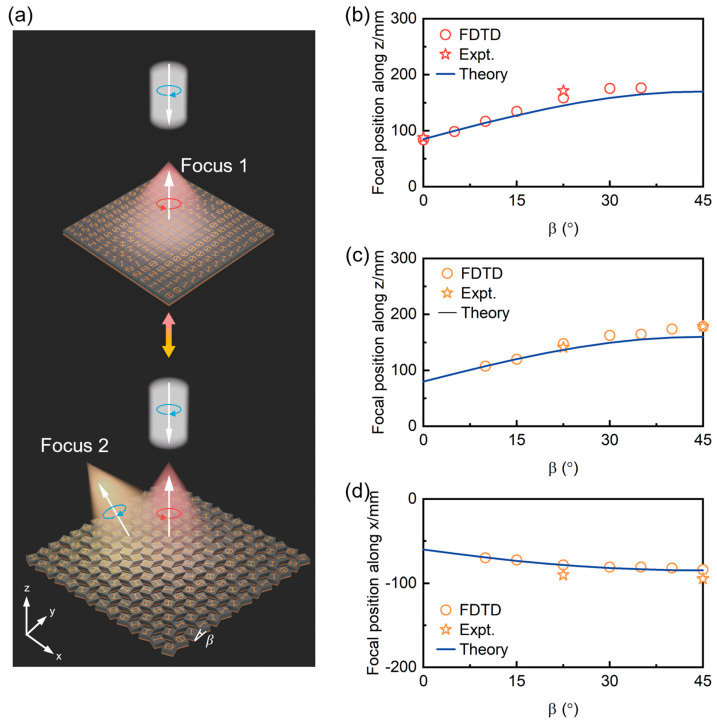
Tunable beam focusing effect by RS kirigami metasurface. (**a**) Schematics of two beam focusing effects by RS kirigami metasurface illuminated by normally incident LCP wave. Here, the red and orange beams originate from the anomalous mode (focus 1) and normal mode (focus 2), respectively. (**b**) Focal position along the z direction of focus 1 as a function of β. (**c**,**d**) Focal position along both the z direction and x direction of focus 2 as a function of β, respectively. Here, the frequency is fixed at 10 GHz.

## Data Availability

Data are contained within the article or Appendix A.

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
