# Peer review of "Electromagnetic Wavefront Engineering by Switchable and Multifunctional Kirigami Metasurfaces"

_nanomaterials, 2025, doi:10.3390/nano15010061_

Round 1
Reviewer 1 Report
Comments and Suggestions for Authors
In this manuscript, Wang et al. theoretically, computationally, and experimentally demonstrate reconfigurable metasurfaces in the microwave regime. In my view, the manuscript introduces a creative concept and is sufficiently rigorous.
I suggest the authors should temper their claim of their devices being “continuously tunable” (Fig. 7). Given the video in the supporting information, it seems that the process of stretching out the metasurfaces is reasonably controllable, but squishing it back together is not at all well-controlled. As far as I can tell, continuous tuning is unrealistic. In addition, could the authors please add a brief discussion about whether the actuation of their metasurface could be improved in the future?
I feel that reconfigurable metasurfaces is a difficult problem in the visible and telecom because of the nanoscale features required at short operating wavelengths. However, the metasurfaces in this manuscript operate in the microwave where reconfigurability is less challenging because feature sizes are larger. While this manuscript introduces an interesting new concept for tunable metasurfaces, its introduction focuses primarily on papers about metasurfaces in the visible/near-infrared, which is not a particularly fair comparison to their devices. The authors should add some discussion of relevant tunable metasurface papers in the microwave (and optionally discuss whether their design can be scaled to shorter wavelengths). A couple examples of tunable microwave metasurfaces (though there are several others if the authors prefer):
· F. Callewaert, V. Velev, S. Jiang, A. V. Sahakian, P. Kumar, and K. Aydin, “Inverse-designed stretchable metalens with tunable focal distance,” Appl. Phys. Lett., vol. 112, no. 9, p. 091102, Feb. 2018, doi: 10.1063/1.5017719.
· Y. Yang, A. Vallecchi, E. Shamonina, C. J. Stevens, and Z. You, “A new class of transformable kirigami metamaterials for reconfigurable electromagnetic systems,” Sci Rep, vol. 13, no. 1, p. 1219, Jan. 2023, doi: 10.1038/s41598-022-27291-8.
The authors may want to increase the size of the labels in most of their plots, as they are currently hard to read.
Comments on the Quality of English LanguageThere are a couple instances where the writing is a bit uncouth, but overall it is understandable enough. For example:
· “Let’s first introduce the fundamental principle” (Avoid contractions in formal writing)
· “undoubtedly confirming the multifunctional and reconfigurable wave modulations based on the proposed RS kirigami metadevices.” (There will always be some room for doubt…)
Author Response
First of all, we would like to deeply thank all reviewers for their time and valuable comments on our manuscript. We have carefully considered all comments raised by the reviewers and revised our manuscript accordingly (changes marked in red color). Below are our detailed responses to the reviewers’ comments.
In this manuscript, Wang et al. theoretically, computationally, and experimentally demonstrate reconfigurable metasurfaces in the microwave regime. In my view, the manuscript introduces a creative concept and is sufficiently rigorous.
Reply: We sincerely thank the reviewer for carefully reviewing our work and providing positive comments such as “a creative concept” and “is sufficiently rigorous”. Below are our detailed responses to all technical comments of the reviewer.
- I suggest the authors should temper their claim of their devices being “continuously tunable” (Fig. 7). Given the video in the supporting information, it seems that the process of stretching out the metasurfaces is reasonably controllable, but squishing it back together is not at all well-controlled. As far as I can tell, continuous tuning is unrealistic. In addition, could the authors please add a brief discussion about whether the actuation of their metasurface could be improved in the future?
Reply: We sincerely thank the reviewer for posing these valuable comments.
Due to the limitations of currently adopted manufacturing and assembly processes, manually tunning kirigami metasurface still faces the challenge of poor continuity. In our current experiment, we first manually apply a global stretching or compression force to the kirigami metasurface and then finely adjust each local rotating unit to ensure the overall uniformity. Such a local fine-tuning process is not illustrated in the movie, which may potentially mislead the readers.
In the future, we may consider combining the kirigami metasurface with a mechanical-control system to improve the accuracy and continuity of the rotational control. Besides, the adoption of 3D printing technique with better accuracy may also improve the uniformity of our device during the transformation process.
Following the suggestion of the reviewer, we have accordingly revised our claims in the revised manuscript (see Page 11 and Page 12):
“We finally investigate the continuous evolution of two beam focusing functionalities.”
“Figure 7. Continuously Tunable beam focusing effect by RS kirigami metasurface.”
- I feel that reconfigurable metasurfaces is a difficult problem in the visible and telecom because of the nanoscale features required at short operating wavelengths. However, the metasurfaces in this manuscript operate in the microwave where reconfigurability is less challenging because feature sizes are larger. While this manuscript introduces an interesting new concept for tunable metasurfaces, its introduction focuses primarily on papers about metasurfaces in the visible/near-infrared, which is not a particularly fair comparison to their devices. The authors should add some discussion of relevant tunable metasurface papers in the microwave (and optionally discuss whether their design can be scaled to shorter wavelengths). A couple examples of tunable microwave metasurfaces (though there are several others if the authors prefer):
- F. Callewaert, V. Velev, S. Jiang, A. V. Sahakian, P. Kumar, and K. Aydin, “Inverse-designed stretchable metalens with tunable focal distance,” Appl. Phys. Lett., vol. 112, no. 9, p. 091102, Feb. 2018, doi: 10.1063/1.5017719.
- Y. Yang, A. Vallecchi, E. Shamonina, C. J. Stevens, and Z. You, “A new class of transformable kirigami metamaterials for reconfigurable electromagnetic systems,” Sci Rep, vol. 13, no. 1, p. 1219, Jan. 2023, doi: 10.1038/s41598-022-27291-8.
Reply: We sincerely thank the reviewer for sharing these valuable references and providing positive comments such as “an interesting new concept for tunable metasurfaces”.
The reviewer is correct that many tunable materials mentioned in the introduction section usually operate in the visible/near-infrared wavelength bands and the kirigami metasurface demonstrated in the microwave regime will face less challenge. However, this technique may be developed to high-frequency regimes. For example, with the advancements in micro- and nano-fabrication techniques, the size of such kirigami-based reconfigurable microstructures can be conveniently scaled down, making them suitable for applications in higher frequencies. Besides, some recent advances also demonstrate the possibility of inducing carefully designed special folds to connect the adjacent meta-atoms in origami-like design [Nature 575, 164 (2019); Science 386, 1031–1037 (2024); Nat. Mater. 2024 https://doi.org/10.1038/s41563-024-02007-7].
To address this issue, we have cited all these suggested references and made the appropriate comments in the manuscript (see Page 2 and Page 13):
“On the other hand, people utilize elastic materials [33-35] or origami/kirigami substrates [36-40] to change the relative spacing between the meta-atoms within the metasurfaces, achieving the dynamical control of transmission characteristics, focal length and holographic image”.
“With the advancements in micro- and nano-fabrication techniques [47-49], we believe that this reconfigurable kirigami technology can be applied to higher frequencies..”
- The authors may want to increase the size of the labels in most of their plots, as they are currently hard to read.
Reply: We sincerely thank the reviewer for providing this important suggestion. To address this issue, we have revised all of our figures in the revised manuscript and Supplementary Materials
- There are a couple instances where the writing is a bit uncouth, but overall it is understandable enough. For example:
- “Let’s first introduce the fundamental principle” (Avoid contractions in formal writing)
- “undoubtedly confirming the multifunctional and reconfigurable wave modulations based on the proposed RS kirigami metadevices.” (There will always be some room for doubt…)
Reply: We sincerely thank the reviewer for posing this important comment. To address this issue, we have accordingly polished our language in the revised version:
- “We first introduce the fundamental principle”
- “undoubtedly verifying the high performance of such a metadevice”
- “undoubtedly confirming the multifunctional and reconfigurable wave modulations based on the proposed RS kirigami metadevices”.
- “During the transformation process, the meta-atoms located in the black and gray regions rotate along in opposite directions in the transformation process, resulting in the a phase difference between them.”
- “Therefore, two anomalous modes are terminated and only the single normal mode survives, corresponding to the case of the third functionality denoted as target 3 in Fig. 1(c).”→ “Consequently, two anomalous modes are terminated, leaving only the single normal mode, which corresponds to the case of the third functionality denoted as target 3 in Fig. 1(c).”
- “As indicated previously, tThe adjacent meta-atoms will rotate along in the opposite directions as described by Eq. (1).”

Reviewer 2 Report
Comments and Suggestions for Authors
The authors present reflective devices that operate at 10 GHz. The devices are mechanically tunable. The device is classified as a metasurface but doesn’t present any functionality to why it is a meta device. The device is also classified as reconfigurable but it luck of reconfigurability since the presented prototypes are fixed to a defined functionalities post their fabrication. Due to this I cannot accept this manuscript for publication.
11) The device in Figure 2 seems to be uniform. Is this a kirigami device? Please clarify this.
22) Section 2.2 and section 2.3 describe coding schemes to achieve different functionalities of with two devices. The derivation of the resonant phase is based on reflect array theory. Which raises the question what ability makes this device a metasurfase?
33) Are the devices multifunctional and reconfigurable or just mechanically tunable? The devices are fixed post-fabrication. Maybe the title of the paper needs to be changed.
44) Is there anything new in the analysis in section 2? The authors need to stress this if there is.
The manuscript describes a kirigami type mechanical surface which can be coded to operate as a mechanically tunable surface. The presented results so far don’t show any new functionality or even ability that can constitute the devices as metadevices.
Author Response
First of all, we would like to deeply thank all reviewers for their time and valuable comments on our manuscript. We have carefully considered all comments raised by the reviewers and revised our manuscript accordingly (changes marked in red color). Below are our detailed responses to the reviewers’ comments.
The authors present reflective devices that operate at 10 GHz. The devices are mechanically tunable. The device is classified as a metasurface but doesn’t present any functionality to why it is a meta device. The device is also classified as reconfigurable but it luck of reconfigurability since the presented prototypes are fixed to a defined functionalities post their fabrication. Due to this I cannot accept this manuscript for publication.
Reply: We sincerely thank the reviewer for providing these comments.
First, according to the definition in the literatures (e.g., Light: Advanced Manufacturing 5, 5 (2024); Photonics Insights 3, R07 (2024)), the so-called metasurfaces are usually composed of artificially designed electromagnetic microstructures with subwavelength scale (often called meta-atoms), enabling functionalities that are challenging or impossible to achieve with traditional optical components. Based on the metasurfaces, numerous novel applications/functionalities are realized, such as anomalous beam deflection, beam focusing, holographic imaging, and so on. For comparison, our metadevices consist of a series of subwavelength artificial microstructures encoded with spatially varying resonant and PB phase profiles, enabling the functionalities of the beam deflection and beam focusing (as shown in Figures. 3 and 6). Therefore, our devices fully agree with the definition of metasurfacess.
Second, we agree with the reviewer that the presented prototypes are fixed to defined functionalities after their fabrication. However, the proposed metasurfaces still exhibit the switchable functionalities among three different choices through the kirigami technique. For example, as described in section 2.3, three different wave-control functionalities including the anomalous reflection, straight beam focusing and tilted beam focusing have been realized in a kirigami metasurface, by simply tuning the transformation state of the device and the polarization of the illumination waves. According to the definition in the literature [e.g., Adv. Sci. 7, 1903382 (2020)], while the tunable metasurfaces usually achieve finely-tuned predefined functionality, the reconfigurable metasurface can possess significantly different functionalities. Therefore, our devices satisfy the definition of the reconfigurable metasurfaces.
To address this issue, we have revised our description in the main text (see Page 5):
“In summary, we can realize the modulation among three different functionalities by controlling the transformation state of the kirigami metasurface and the illumination of different CP waves.”
- The device in Figure 2 seems to be uniform. Is this a kirigami device? Please clarify this.
Reply: We sincerely thank the reviewer for posing this question. The device shown in Figure 2 is not the kirigami metausrface but the periodic array of the designed meta-atoms which are the building block for constructing the kirigami device. We should first clarify the fundamental electromagnetic properties of the meta-atoms (including the resonant phase and polarization conversion ratio), which will determine the performance of the kirigami metasurface. The two kirigami devices can be referred to Figure 3 and Figure 5, respectively.
To fully address this comment and prevent the possible misleading information, we have revised our description in the main text (see Page 5):
“Figure 2. Design of high-efficiency composite-phase meta-atoms for the desired metasurface.”
- Section 2.2 and section 2.3 describe coding schemes to achieve different functionalities of with two devices. The derivation of the resonant phase is based on reflect array theory. Which raises the question what ability makes this device a metasurfase?
Reply: We thank the reviewer for posing this question. The reviewer is correct that the resonant phase is obtained based on the reflect array theory. In our design process, we first calculate the reflection properties (including the resonant phase and polarization conversion ratio (PCR)) of the meta-atoms in a periodic array (see Figure 2). Then, we can construct the kirigami composite-phase metasurface by integrating a series of meta-atoms with different orientation angles and geometry sizes. Such a design strategy is widely utilized in the community (see Refs: Phys. Rev. Lett. 118,113901 (2017); Nano Lett. 19, 1158-1165 (2019); Adv. Sci. 10, 2205499 (2023)). We have adopted the same strategy to design our metasurfaces as shown in Sec. 2.2 and 2.3. Compared to the previous studies, this work proposed to introduce the kirigami technique into the design of composite-phase metasurfaces. Via changing the polarization state of input waves and the transformation state of the metasurface, we can flexibly change the functionalities of our kirigami metasurfaces in three different choices, enabling the reconfigurable properties.
- Are the devices multifunctional and reconfigurable or just mechanically tunable? The devices are fixed post-fabrication. Maybe the title of the paper needs to be changed.
Reply: We sincerely thank the reviewer for providing this question. The proposed kirigami metasurfaces can indeed achieve the multifunctional and reconfigurable properties. The conventional tunable metasurfaces can only exhibit one predefined tunable functionality, e.g., beam deflection or focusing (see Refs: Nano Lett. 16, 2818−2823 (2016); Adv. Funct. Mater. 32, 2107699 (2022); Nat. Commun. 9, 812 (2018)). For comparison, the proposed kirigami metasurface can achieve three different functionalities by changing its transformation state and the polarization state of the input waves. For example, the first kirigami metasurfce shown in Figure. 3 can achieve three beam deflection modes with different deflection angles. The second kirigami metasurface shown in Figure 5 can achieve another three functionalities, including straight beam focusing, tilted beam focusing and anomalous reflection.
To address this comment, we added some comments on this issue in Page 5:
“In summary, we can realize the modulation among three different functionalities by controlling the transformation state of the kirigami metasurface and the illumination of different CP waves.”
- Is there anything new in the analysis in section 2? The authors need to stress this if there is.
Reply: We sincerely thank the reviewer for posing this question.
In section 2.1, we introduce the working principle of dynamically controllable metasurface based on kirigami technique and the practical design of high-efficiency meta-atoms for constructing the desired metasurfaces. Then, as a proof of concept, we design a kirigami metasurface that successfully demonstrates dynamic controls of three-channel beam steering in section 2.2. To further demonstrate the potential applications of the proposed concept, we construct another kirigami metasurface that can realize three functionalities including straight beam focusing, tilted beam focusing, and anomalous reflection in section 2.3.
Compared to the previous tunable metasurfaces, the proposed metasurface based on kirigami technique is a new direction in this field. While the conventional mechanical metasurfaces can achieve one tunable functionalities (beam deflection or focusing) by simply tuning the lattice constant, the present kirigami metadevices encoded with both resonant and PB phases can achieve the multifunctional and reconfigurable functionalities by changing the lattice constant and local phase during the transformation process. Three different functionalities can be achieved in the single kirigami metasurface by tuning its transformation state under the illumination of different circular polarization (CP) waves.
To address this comment, we have added proper comments in the main text (see Page 3):
“Such a rotation operation can modulate the orientations of the meta-atoms, thereby synchronously altering the encoded phase profile and the lattice constant of the meta-atoms.”
- The manuscript describes a kirigami type mechanical surface which can be coded to operate as a mechanically tunable surface. The presented results so far don’t show any new functionality or even ability that can constitute the devices as metadevices.
Reply: We sincerely thank the reviewer for providing this comment, which are similar to the previous ones. Here, we summarize our response and explanation.
The proposed metadevice, composed of subwavelength microstructures with spatially varied geometric sizes and orientation angles, can exhibit several different wave-control functionalities, including beam deflection and beam focusing. The multifunctional and reconfigurable properties are enabled by the mechanical-control technique of kirigami transformation and the design strategy of composite-phase coding (i.e., resonant phase and PB phase). These properties are not available in the conventional mechanically tunable metasurfaces (see Refs: Adv. Funct. Mater. 32, 2107699 (2022); Nat. Commun. 9, 812 (2018)). Both the design strategy and the achieved functionalities fully match the definition of the metadevices in the community. We believe that the present results open up a new direction for achieving reconfigurable EM wavefront engineering based on the multiple degrees of freedom.

Round 2
Reviewer 1 Report
Comments and Suggestions for Authors
The authors have addressed my concerns. I suggest the editors accept the manuscript.
Author Response
The authors have addressed my concerns. I suggest the editors accept the manuscript.
Reply: Thank you very much for your supportive comments. We are delighted that our revisions have fully addressed your concerns and that you recommend our manuscript for acceptance.
Reviewer 2 Report
Comments and Suggestions for Authors
My comments have been partially addressed but the work has improved. Please find below a continuation to my list of comments.
Please show in your response also the actions and the location of your corrections as to aid with the review process.
1) This has been clarified.
2) I am satisfied with the response.
3) I am not satisfied with the authors’ response. If the prototype is fixed post fabrication then it is not reconfigurable. I would suggest to find a more appropriate title this device is “multifunctional and switchable” as you mentioned in your response.
4) I am satisfied with the authors response.
5) Based on the authors response the main novelty of the work is the multifunctional demonstration with this new type kirigami metasurface . Since they references other kirigami implementations I would strongly recommend to emphasize this and to compare their advantage over previous work with kirigami surfaces. Also mention more alternative implementations since a multifunctional behavior has been showed with other implementations. Once you have shown this then conclude with the advantages of the kirigami implementation in a particular application.
Although alternative implementations are mentioned in the introduction some worth mentioning include active meta-atoms for cloaking, other materials like Liquid crystals, graphene, amplitude and phase control with PIN diodes and ICs for arbitrary wavefront reflection.
[a] Selvanayagam, M., & Eleftheriades, G. v. (2013). Experimental Demonstration of Active Electromagnetic Cloaking. Physical Review X, 3(4), 041011
[b] A. Fallahi and J. Perruisseau-Carrier, “Design of tunable biperiodic graphene metasurfaces,” Phys. Rev. B - Condens. Matter Mater. Phys., vol. 86, no. 19, pp. 1–9, 2012.
[c] A. D. Squires et al., “Electrically tuneable terahertz metasurface enabled by a graphene/gold bilayer structure,” Commun. Mater., vol. 3, no. 1, p. 56, Aug. 2022.
[d] K. M. Kossifos, J. Georgiou and M. A. Antoniades, "ASIC-Enabled Programmable Metasurfaces—Part 2: Performance and Synthesis," in IEEE Transactions on Antennas and Propagation, vol. 72, no. 3, pp. 2800-2810, March 2024, doi: 10.1109/TAP.2024.3349771
[e] K. M. Kossifos, J. Georgiou and M. A. Antoniades, "ASIC-Enabled Programmable Metasurfaces—Part 1: Design and Characterization," in IEEE Transactions on Antennas and Propagation, vol. 72, no. 3, pp. 2790-2799, March 2024, doi: 10.1109/TAP.2024.3349685
[f] M. K. Emara, D. Kundu, K. Macdonell, L. Rufail and S. Gupta, "Reconfigurable Metasurface Reflectors Using Split-Ring Resonators With Co-Designed Biasing for Magnitude/Phase Control," in IEEE Transactions on Antennas and Propagation, vol. 72, no. 9, pp. 7425-7430, Sept. 2024, doi: 10.1109/TAP.2024.3428729
Other mechanical or mechatronic metasurfaces can be added in the introduction to aid with emphasizing your advantage.
Author Response
My comments have been partially addressed but the work has improved. Please find below a continuation to my list of comments. Please show in your response also the actions and the location of your corrections as to aid with the review process.
Reply: We sincerely thank the reviewer for accepting our responses to most of the comments in the last report. Below are our detailed responses to the additional technical comments.
- I am not satisfied with the authors’ response. If the prototype is fixed post fabrication then it is not reconfigurable. I would suggest to find a more appropriate title this device is “multifunctional and switchable” as you mentioned in your response.
Reply: We thank the reviewer for providing this suggestion and have replaced the description of “reconfigurable” in the manuscript with “switchable”.
- Based on the authors response the main novelty of the work is the multifunctional demonstration with this new type kirigami metasurface . Since they references other kirigami implementations I would strongly recommend to emphasize this and to compare their advantage over previous work with kirigami surfaces. Also mention more alternative implementations since a multifunctional behavior has been showed with other implementations. Once you have shown this then conclude with the advantages of the kirigami implementation in a particular application.
Although alternative implementations are mentioned in the introduction some worth mentioning include active meta-atoms for cloaking, other materials like Liquid crystals, graphene, amplitude and phase control with PIN diodes and ICs for arbitrary wavefront reflection.
[a] Selvanayagam, M., & Eleftheriades, G. v. (2013). Experimental Demonstration of Active Electromagnetic Cloaking. Physical Review X, 3(4), 041011
[b] A. Fallahi and J. Perruisseau-Carrier, “Design of tunable biperiodic graphene metasurfaces,” Phys. Rev. B - Condens. Matter Mater. Phys., vol. 86, no. 19, pp. 1–9, 2012.
[c] A. D. Squires et al., “Electrically tunable terahertz metasurface enabled by a graphene/gold bilayer structure,” Commun. Mater., vol. 3, no. 1, p. 56, Aug. 2022.
[d] K. M. Kossifos, J. Georgiou and M. A. Antoniades, "ASIC-Enabled Programmable Metasurfaces—Part 2: Performance and Synthesis," in IEEE Transactions on Antennas and Propagation, vol. 72, no. 3, pp. 2800-2810, March 2024, doi: 10.1109/TAP.2024.3349771
[e] K. M. Kossifos, J. Georgiou and M. A. Antoniades, "ASIC-Enabled Programmable Metasurfaces—Part 1: Design and Characterization," in IEEE Transactions on Antennas and Propagation, vol. 72, no. 3, pp. 2790-2799, March 2024, doi: 10.1109/TAP.2024.3349685
[f] M. K. Emara, D. Kundu, K. Macdonell, L. Rufail and S. Gupta, "Reconfigurable Metasurface Reflectors Using Split-Ring Resonators With Co-Designed Biasing for Magnitude/Phase Control," in IEEE Transactions on Antennas and Propagation, vol. 72, no. 9, pp. 7425-7430, Sept. 2024, doi: 10.1109/TAP.2024.3428729
Other mechanical or mechatronic metasurfaces can be added in the introduction to aid with emphasizing your advantage.
Reply: We sincerely thank the reviewer for posing these valuable comments and providing these references.
First, the reviewer is correct that a comparison should be made between the conventional kirigami metasruface and the proposed system. As discussed in the original manuscript, people have reported some origami/kirigami metasurfaces in which only the relative spacing between the meta-atoms can be adjusted. As a result, the achieved functionality can be just simply tuned, e.g., the change of focal length or deflection angle. In contrast, the “rotating square” kirigami technique proposed in the manuscript is able to synchronously tailor the lattice constant and local phase of the meta-atoms, thus achieving tunable and switchable multi-functionalities (e.g., the change from beam bending to beam focusing in the 2nd kirigami device).
To address this comment, we have accordingly added more comparison in the revised manuscript (see Page 12):
“Different from previous origami/kirigami technique, this scheme enables the flexible modulations of the lattice constant as well as phase profile of the kirigami metasurface.”
Second, as indicated by the reviewer, in the introduction part of the original manuscript we have discussed the tunable/reconfigurable metasurfaces integrated with different active materials, including liquid crystals, 2D materials, PIN diodes and varactors. All these suggested references can be classified as the electrically-controlled tunable meta-devices. While this tuning mechanism allows the flexible modulation of arbitrary wavefronts with a fast response speed, it still suffer from the challenges of complicated control-system and high cost. As a new multifunctional tuning method for dynamic metadevices, kirigami technique has the advantages of simple control-system and low cost.
To address this issue, we have cited the suggested references in the introduction section and made the appropriate comments in the manuscript (see Page 2):
“For instance, researchers have utilized semiconductors [12], phase change materials [13-15] or 2D materials [16-19] to effectively modulate the scattering field amplitude or phase of meta-atoms by altering their material properties in response to an applied electrical [16, 18], thermal [14, 19], optical [20, 21] or chemical [22, 23] stimulus. While most tuning strategies are utilized for the flexible change of the fundamental EM parameters based on homogeneous metamaterials, they are usually limited by the tuning range or specific operating frequency bands. Typically, via integrating electronic components such as PIN diodes and varactors with the local meta-atoms, researchers have constructed homogeneous or gradient metasurfaces for achieving dynamic wavefront engineering including arbitrary polarization conversion [24, 25], beam steering [26-32], active EM cloaking [33] and dynamic holography [34]. These methods show the advantages of quick response and multiple functionalities, but still suffer from the problems of complicated control-system and high cost. Therefore, a flexible-control and low-cost approach to achieve tunable/switchable wavefront control is highly desired.”
